# Meta-Calibration: Learning of Model Calibration Using Differentiable Expected Calibration Error

**Ondrej Bohdal**                                                              *ondrej.bohdal@ed.ac.uk*
*The University of Edinburgh*

**Yongxin Yang**                                                              *yongxin.yang@qmul.ac.uk*
*Queen Mary University of London*

**Timothy Hospedales**                                                        *t.hospedales@ed.ac.uk*
*The University of Edinburgh*
*Samsung AI Center Cambridge*

**Reviewed on OpenReview:** *https://openreview.net/forum?id=R2hUure38l*

## Abstract

Calibration of neural networks is a topical problem that is becoming more and more important as neural networks increasingly underpin real-world applications. The problem is especially noticeable when using modern neural networks, for which there is a significant difference between the confidence of the model and the probability of correct prediction. Various strategies have been proposed to improve calibration, yet accurate calibration remains challenging. We propose a novel framework with two contributions: introducing a new differentiable surrogate for expected calibration error (DECE) that allows calibration quality to be directly optimised, and a meta-learning framework that uses DECE to optimise for validation set calibration with respect to model hyper-parameters. The results show that we achieve competitive performance with existing calibration approaches. Our framework opens up a new avenue and toolset for tackling calibration, which we believe will inspire further work on this important challenge.

## 1 Introduction

When deploying neural networks to real-world applications, it is crucial that models' own confidence estimates accurately match their probability of making a correct prediction. If a model is over-confident about its predictions, we cannot rely on it; while well-calibrated models can abstain or ask for human feedback in the case of uncertain predictions. Models with accurate confidence estimates about their own predictions can be described as well-calibrated. This is particularly important in applications involving safety or human impact – such as autonomous vehicles (Bojarski et al., 2016; Wiseman, 2022) and medical diagnosis (Jiang et al., 2012; Caruana et al., 2015; El-Sappagh et al., 2023), and tasks that directly rely on calibration such as outlier detection (Hendrycks & Gimpel, 2017; Liang et al., 2018; Wang et al., 2023). However, modern neural networks are known to be badly calibrated (Guo et al., 2017; Gawlikowski et al., 2021).

This challenge of calibrating neural networks has motivated a growing area of research. Perhaps the simplest approach is to post-process predictions with techniques such as temperature scaling (Guo et al., 2017). However, this has limited efficacy (Wang et al., 2021) and fails in the common situation of distribution shift between training and testing data (Ovadia et al., 2019; Tomani et al., 2021). It also reduces network's confidence in correct predictions. Another family of approaches modifies the model training regime to improve calibration. Müller et al. (2019) show that label smoothing regularization improves calibration by increasing overall predictive entropy. But it is unclear how to set the label smoothing parameter so as to optimise calibration. Mukhoti et al. (2020) show that Focal loss leads to better calibrated models than standard

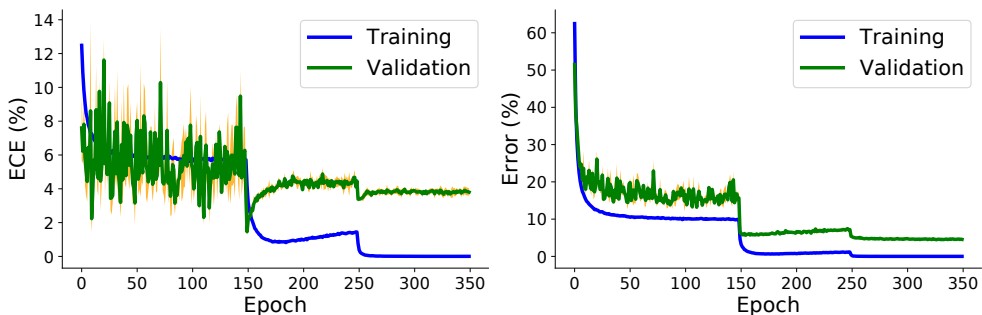

Figure 1: Illustration of the calibration generalisation gap. ECE and classification error during training of ResNet18 on CIFAR-10, using cross-entropy loss and showing mean and std. across three random seeds. Training ECE and error fall to 0. However, calibration overfitting occurs and validation ECE increases. This motivates the need for meta-learning to tune hyper-parameters to optimise *validation* calibration.

cross-entropy, and Kumar et al. (2018) introduce a proxy for calibration error to be minimised along with standard cross-entropy on the training set. However, this does not ensure calibration on the test set. Ovadia et al. (2019); Mosser & Naeini (2022) show that Bayesian neural network approaches are comparatively well-calibrated, however these are difficult and expensive to scale to large modern architectures.

The above approaches explore the impact of various architectures and design parameters on calibration. In this paper we step back and consider how to optimise for calibration. Direct optimisation for calibration would require a differentiable metric for calibration. However, calibration is typically measured using expected calibration error (ECE), which is not differentiable due to its internal binning/counting operation. Therefore our first contribution is a high-quality differentiable approximation to ECE, which we denote DECE. A second consideration is how to optimise for calibration – given that calibration itself is a quantity with a generalisation gap between training and validation (Carrell et al., 2022). We illustrate this challenge in Figure 1, which shows how validation calibration worsens as training calibration improves during training. To this end, our second contribution is to introduce a framework for meta-learning model calibration: We fit a model on the training set using a given set of hyper-parameters, evaluate it on a disjoint validation set, and optimise for the hyper-parameters that lead to the best *validation* calibration as measured by DECE. Our framework for differentiable optimisation of validation calibration is generic and can potentially be used with any continuous model hyper-parameters. Our third contribution is a specific choice of hyper-parameters which, when meta-learned with a suitable calibration objective, are effective for tuning the base model's calibration. Specifically, we propose non-uniform label-smoothing, which can be tuned by meta-learning to penalise differently each unique combination of true and predicted label.

To summarize, we present a novel framework and toolset for improving model calibration by differentiable optimisation of model hyper-parameters with respect to validation calibration. We analyse our differentiable calibration metric in detail, and show that it closely matches the original non-differentiable metric. When instantiated with label smoothing hyper-parameters, our empirical results show that our framework produces high-accuracy and well-calibrated models that are competitive with existing methods across a range of benchmarks and architectures.

## 2 Related work

**Calibration**   Since finding modern neural networks are typically miscalibrated (Guo et al., 2017), model calibration has become a popular area for research (Gawlikowski et al., 2021) with many applications, including medical image segmentation (Judge et al., 2022) and object detection (Munir et al., 2023). Guo et al. (2017) study a variety of potential solutions and find simple post-training rescaling of the logits – temperature scaling – works relatively well. Kumar et al. (2018) propose a kernel-based measure of calibration called MMCE that they use as regularization during training of neural networks. Mukhoti et al. (2020) show Focal loss – a

relatively simple weighted alternative to cross-entropy – can be used to train well-calibrated neural networks. The classic Brier score (Brier, 1950), which is the squared error between the softmax vector with probabilities and the ground-truth one-hot encoding, has also been shown to work well. Similarly, label smoothing (Müller et al., 2019) has been shown to improve model calibration. Noise during training more broadly can also be beneficial for improving calibration (Ferianc et al., 2023). These aforementioned methods do not optimise for calibration metric (e.g., ECE) directly, because the calibration metrics are usually non-differentiable. In this work, we propose a new high-quality differentiable approximation to ECE, and utilize it with meta-learning.

Karandikar et al. (2021) have proposed soft-binned ECE (SB-ECE) as an auxiliary loss to be used during training to encourage better calibration. The approach makes the binning operation used in ECE differentiable, leading to an additional objective that is more compatible with gradient-based methods. SB-ECE does not make the accuracy component of ECE differentiable, but we make all components of ECE differentiable. We also try to obtain a highly accurate approximation of the binning operation, while SB-ECE binning estimate for the left-most and right-most bin can be inaccurate as a result of using bin's center value. We provide additional comparison with SB-ECE in the appendix, showing DECE provides a closer approximation to ECE than SB-ECE, and that empirically our meta-learning approach with DECE leads to better calibration.

**Meta-learning**  We use the newly introduced DECE metric as part of the meta-objective for gradient-based meta-learning. Gradient-based meta-learning has become popular since the seminal work MAML (Finn et al., 2017) has successfully used it to solve the challenging problem of few-shot learning (Wang et al., 2020; Song et al., 2023; Bohdal et al., 2023). Nevertheless, gradient-based meta-learning is not limited to few-shot learning problems, but it can also be used to solve various other challenges, including training with noisy labels (Shu et al., 2019; Algan & Ulusoy, 2022), dataset distillation (Wang et al., 2018; Bohdal et al., 2020; Yu et al., 2023), domain generalization and adaptation (Li et al., 2019; Balaji et al., 2018; Bohdal et al., 2022), molecular property prediction (Chen et al., 2023) and many others.

Gradient-based meta-learning is typically formulated as a bilevel optimisation problem where the main model is trained in the inner loop and the meta-knowledge or hyper-parameters are trained in the outer loop. In the case of few-shot learning it is possible to fully train the model within the inner loop – also known as offline meta-learning. In more realistic and larger scale settings such as ours, it is only feasible to do one or a few updates in the inner loop. This approach is known as online meta-learning (Hospedales et al., 2021) and means that we jointly train the main model as well as the meta-knowledge. Online meta-learning is most commonly done using the so-called $T_1 - T_2$ method (Luketina et al., 2016) that updates the meta-knowledge by backpropagating through one step of the main model update. This is the approach that we adopt, however more advanced or efficient approaches are also available (Lorraine et al., 2020; Bohdal et al., 2021).

**Label smoothing**  We use label smoothing as the meta-knowledge that we use to demonstrate the benefits of using our DECE metric. Label smoothing has been proposed by Szegedy et al. (2016) as a technique to alleviate overfitting and improve the generalization of neural networks. It consists of replacing the one-hot labels by their softer alternative that gives non-zero target probabilities to the incorrect classes. Label smoothing has been studied in more detail, for example Müller et al. (2019) have observed that label smoothing can improve calibration, but at the same time it can hurt knowledge distillation if used for training the teacher. Mukhoti et al. (2020) have compared Focal loss with label smoothing among other approaches, showing that simple label smoothing strategy has a limited scope for state-of-the-art calibration.

However, we demonstrate that using meta-learning and our DECE objective, a more expressive form of label smoothing can achieve state-of-the-art calibration results. Note that meta-learning has already been used for label smoothing (Li et al., 2020), but using it as meta-knowledge to directly optimise calibration is new.

## 3 Methods

### 3.1 Preliminaries

We first discuss expected calibration error (ECE) (Naeini et al., 2015), before we derive a differentiable approximation to it. ECE measures the expected difference (in absolute value) between the accuracies and

the confidences of the model on examples that belong to different confidence intervals. ECE is defined as

$$\text{ECE} = \sum_{m=1}^{M} \frac{|B_m|}{n} \left| \text{acc}\left(B_m\right) - \text{conf}\left(B_m\right) \right|,$$

where accuracy and confidence for bin $B_m$ are

$$\text{acc}\left(B_m\right) = \frac{1}{|B_m|} \sum_{i \in B_m} \mathbf{1}\left(\hat{y}_i = y_i\right)$$

$$\text{conf}\left(B_m\right) = \frac{1}{|B_m|} \sum_{i \in B_m} \hat{p}_i.$$

There are $M$ interval bins each of size $1/M$ and $n$ samples. Confidence $\hat{p}_i$ is the probability of the top prediction as given by the model for example $i$. We group the confidences into their corresponding bins, with bin $B_m$ covering interval $(\frac{m-1}{M}, \frac{m}{M}]$. The predicted class of example $i$ is $\hat{y}_i$, while $y_i$ is the true class of example $i$ and $\mathbf{1}$ is an indicator function.

ECE metric is not differentiable because assigning examples into bins is not differentiable and also accuracy is not differentiable due to the indicator function. We propose approximations to both binning and accuracy and derive a new metric called differentiable ECE (DECE).

### 3.2 Differentiable ECE

ECE is composed of accuracy, confidence and binning, but only confidence is differentiable.

**Differentiable accuracy** In order to obtain a differentiable approximation to accuracy, we consider approaches that allow us to find a differentiable way to calculate the rank of a given class. Two approaches stand out: differentiable ranking (Blondel et al., 2020) and an all-pairs approach (Qin et al., 2010). While both allow us to approximate the rank in a differentiable way, differentiable ranking is implemented on CPU only, which would introduce a potential bottleneck for modern applications. All-pairs approach has asymptotic complexity of $\mathcal{O}(n^2)$ for $n$ classes, while differentiable ranking is $\mathcal{O}(n \log n)$. However, if the number of classes is not in thousands or millions, differentiable ranking would be slower because of not using GPUs. We use the all-pairs approach to estimate the rank of a given class.

All-pairs (Qin et al., 2010) calculates a rank of class $i$ as $[R\left(\cdot\right)]_i = 1 + \sum_{j \neq i} \mathbf{1}\left[\phi_i < \phi_j\right]$, where $\phi$ are the logits. We can obtain soft ranks by replacing the indicator function with a sigmoid scaled with some temperature value $\tau_a$ to obtain reliable estimates of the rank of the top predicted class. Once the rank $[R(\cdot)]_l$ for true class $l$ is calculated, we can estimate the accuracy as $\text{acc} = \max(0, 2 - [R]_l)$.

**Soft binning** Our approach is similar to (Yang et al., 2018). We take confidence $\hat{p}_i$ for example $x_i$ and pass it through one-layer neural network $\text{softmax}((w\hat{p}_i + b)/\tau_b)$ parameterized with different values of $w$ and $b$ as explained in (Yang et al., 2018), with temperature $\tau_b$ to control the binning. This leads to $M$ different probabilities, saying how likely it is that $\hat{p}_i$ belongs to the specific bin $B_{m \in 1..M}$. We will denote these probabilities as $o_m(x_i) = p(B_m|\hat{p}_i)$.

Putting these parts together, we define DECE using a minibatch of $n$ examples as:

$$\text{DECE} = \sum_{m=1}^{M} \frac{\sum_{i=1}^{n} o_m(x_i)}{n} \left| \text{acc}\left(B_m\right) - \text{conf}\left(B_m\right) \right|,$$

$$\text{acc}\left(B_m\right) = \frac{1}{\sum_{i=1}^{n} o_m(x_i)} \sum_{i=1}^{n} o_m(x_i) \mathbf{1}\left(\hat{y}_i = y_i\right),$$

$$\text{conf}\left(B_m\right) = \frac{1}{\sum_{i=1}^{n} o_m(x_i)} \sum_{i=1}^{n} o_m(x_i) \hat{p}_i.$$

### 3.3 Meta-learning

Differentiable ECE provides an objective to optimise, but we still need to decide how to utilize it. One option could be to directly use it as an extra objective in combination with standard cross-entropy, as used by a few existing attempts (Karandikar et al., 2021; Kumar et al., 2018). However, we expect this to be unhelpful as calibration on the *training* set is usually good – the issue being a failure of calibration generalisation to the held out validation or test set (Carrell et al., 2022), as illustrated in Figure 1. Meanwhile multi-task training with such non-standard losses may negatively affect the learning dynamics of existing well tuned model training procedures. To optimise for calibration of held-out data, without disturbing standard model training dynamics, we explore the novel approach of using DECE as part of the objective for hyper-parameter meta-learning in an outer loop that wraps an inner learning process of conventional cross-entropy-driven model training.

**Meta-learning objective** We formulate our approach as a bilevel optimisation problem. Our model is assumed to be composed of feature extractor $\boldsymbol{\theta}$ and classifier $\boldsymbol{\phi}$. These are trained to minimise $\mathcal{L}_{CE}^{train}$, cross-entropy loss on the training set. The goal of meta-learning is to find hyper-parameters $\boldsymbol{\omega}$ so that training with them optimises the meta-objective computed on the meta-validation set. In our case the meta-objective is a combination of cross-entropy and DECE to reflect that the meta-learned hyper-parameters should lead to both good generalization and calibration. More specifically the meta-objective is $\mathcal{L}_{CE+\lambda DECE}^{val}$, with hyper-parameter $\lambda$ specifying how much weight is placed on calibration. The bilevel optimisation problem can then be summarized as:

$$\boldsymbol{\omega}^* = \arg\min_{\boldsymbol{\omega}} \mathcal{L}_{CE+\lambda DECE}^{val}(\boldsymbol{\phi}^* \circ \boldsymbol{\theta}^*(\boldsymbol{\omega})),$$
$$\boldsymbol{\phi}^*, \boldsymbol{\theta}^*(\boldsymbol{\omega}) = \arg\min_{\boldsymbol{\phi},\boldsymbol{\theta}} \mathcal{L}_{CE}^{train}(\boldsymbol{\phi} \circ \boldsymbol{\theta}, \boldsymbol{\omega}). \tag{1}$$

To solve this efficiently, we adopt online meta-learning approach (Luketina et al., 2016; Hospedales et al., 2021) where we alternate base model and hyper-parameter updates. This is an efficient strategy as we do not need to backpropagate through many inner-loop steps or retrain the model from scratch for each update of meta-knowledge.

When simulating training during the inner loop, we only update the classifier and keep the feature extractor frozen for efficiency, as suggested by (Balaji et al., 2018). Base model training is done separately using a full model update and a more advanced optimiser.

We give the overview of our meta-learning algorithm in Algorithm 1. The inner loop that trains the main model $(\boldsymbol{\theta}, \boldsymbol{\phi})$ (line 10) is conducted using hyper-parameters $\boldsymbol{\omega}$, while the outer loop (line 12) that trains the hyper-parameters does not directly use it for evaluating the meta-objective (e.g. no learnable label smoothing is applied to the meta-validation labels that are used in the outer loop). We backpropagate through one step of update of the main model.

**Hyper-parameter choice** A key part of meta-learning is to select suitable meta-knowledge (hyper-parameters) that we will optimise to achieve the meta-learning goal (Hospedales et al., 2021). Having cast calibration optimisation as a meta-learning process, we are free to use any of the wide range of hyper-parameters surveyed in (Hospedales et al., 2021). Note that in contrast to grid search that standard temperature scaling (Guo et al., 2017; Mukhoti et al., 2020) and other approaches rely on, we have gradients with respect to hyper-parameters and so can therefore potentially optimise calibration with respect to high-dimensional hyper-parameters. In this paper we explore two options to demonstrate this generality: 1) Unit-wise L2 regularization coefficients of each weight in the classifier layer, inspired by (Balaji et al., 2018) and (Lorraine et al., 2020); and 2) various types of learnable label smoothing (LS) (Müller et al., 2019). However, we found LS to be better overall, so our experiments focus on this and selectively compare against L2.

**Learnable label-smoothing** Learnable label smoothing learns one or more coefficients to smooth the one-hot encoded labels. More formally, if there are $K$ classes in total, $y_k$ is 1 for the correct class $k = c$ and

---

**Algorithm 1** Meta-Calibration

1: **Input:** $\alpha$, $\beta$: inner and outer-loop learning rates
2: **Output:** trained feature extractor $\boldsymbol{\theta}$, classifier $\boldsymbol{\phi}$ and label smoothing $\boldsymbol{\omega}$
3: $\boldsymbol{\omega} \sim p(\boldsymbol{\omega})$
4: $\boldsymbol{\phi}, \boldsymbol{\theta} \sim p(\boldsymbol{\phi}), p(\boldsymbol{\theta})$
5: **while** training **do**
6:     Sample minibatch of training $x_t, y_t$ and meta-validation $x_v, y_v$ examples
7:     **// For LS: use $\boldsymbol{\omega}$ to smooth $y_t$**
8:     **// For L2: add unit-wise weight-decay $\boldsymbol{\omega}$**
9:     Calculate $\mathcal{L}_i = \mathcal{L}_{CE}\left(f_{\boldsymbol{\phi}\circ\boldsymbol{\theta}}\left(x_t\right), y_t, \boldsymbol{\omega}\right)$
10:    Update $\boldsymbol{\theta}, \boldsymbol{\phi} \leftarrow \boldsymbol{\theta}, \boldsymbol{\phi} - \alpha\nabla_{\boldsymbol{\theta},\boldsymbol{\phi}}\mathcal{L}_i$
11:    Calculate $\mathcal{L}_o = \mathcal{L}_{CE+\lambda DECE}\left(f_{\boldsymbol{\phi}\circ\boldsymbol{\theta}}\left(x_v\right), y_v\right)$
12:    Update $\boldsymbol{\omega} \leftarrow \boldsymbol{\omega} - \beta\nabla_{\boldsymbol{\omega}}\mathcal{L}_o$
13: **end while**

---

$y_k$ is 0 for all classes $k \neq c$, then with learnable label smoothing $\boldsymbol{\omega}$ the soft label for class $k$ becomes

$$y_k^{LS} = y_k(1 - \omega_{c,k}) + \omega_{c,k}/K.$$

In the scalar case of label smoothing, $\omega_c$ is the same for all classes, while for the vector case it takes different values for each class $c$. We consider scalar and vector variations as part of ablation.

Our main variation of meta-calibration uses non-uniform label smoothing. It is computed using the overall strength of smoothing $\omega_c^s$ for the correct class $c$ and also $\omega_{c,k}^d$ saying how $\omega_c^s$ is distributed across the various incorrect classes $k \neq c$. Given correct class $c$, with this variation the soft label for class $k$ is calculated as:

$$y_k^{LS} = y_k(1 - \omega_c^s) + \omega_c^s \frac{\omega_{c,k}^d}{\epsilon + \sum_{i=1}^{K} \omega_{c,i}^d},$$

where we normalize the distribution weights to sum to 1 and use small value $\epsilon$ to avoid division by 0. The learnable label smoothing parameters are restricted to non-negative values, with the total label smoothing strength at most 0.5. In practice the above is implemented by learning a vector of $K$ elements specifying the strengths of overall label smoothing for different correct classes $c$, and a matrix of $K \times (K-1)$ elements specifying how the label smoothing is distributed across the incorrect classes $k \neq c$.

**Learnable L2 regularization** In the case of learnable L2 regularization (cf: Balaji et al. (2018)), the goal is to find unit-wise L2 regularization coefficients $\boldsymbol{\omega}$ for the classifier layer $\boldsymbol{\phi}$ so that training with them optimises the meta-objective that includes DECE ($\boldsymbol{\theta}$ is the feature extractor). The inner loop loss becomes

$$\mathcal{L}_i = \mathcal{L}_{CE}\left(f_{\boldsymbol{\phi}\circ\boldsymbol{\theta}}\left(x_t\right), y_t\right) + \boldsymbol{\omega}\|\boldsymbol{\phi}\|^2.$$

## 4 Experiments

Our experiments show that DECE-driven meta-learning can be used to obtain excellent calibration across a variety of benchmarks and models.

### 4.1 Calibration experiments

**Datasets and settings** We experiment with CIFAR-10 and CIFAR-100 benchmarks (Krizhevsky, 2009), SVHN (Netzer et al., 2011) and 20 Newsgroups dataset (Lang, 1995), covering both computer vision and NLP. For CIFAR benchmark, we use ResNet18, ResNet50, ResNet110 (He et al., 2015) and WideResNet26-10 (Zagoruyko & Komodakis, 2016) models. For SVHN we use ResNet18, while for 20 Newsgroups we use global pooling CNN (Lin et al., 2014). We extend the implementation provided by (Mukhoti et al., 2020) to

implement and evaluate our meta-learning approach. We use the same hyper-parameters as selected by the authors for fair comparison, which we summarize next.

CIFAR and SVHN models are trained for 350 epochs, with a multi-step scheduler that decreases the initial learning rate of 0.1 by a factor of 10 after 150 and 250 epochs. Each model is trained with SGD with momentum of 0.9, weight decay of 0.0005 and minibatch size of 128. 90% of the original training set is used for training and 10% for validation. In the case of meta-learning, we create a further separate *meta-validation* set that is of size 10% of the original training data, so we directly train with 80% of the original training data. 20 Newsgroups models are trained with Adam optimiser with the default parameters, 128 minibatch size and for 50 epochs. As the final model we select the checkpoint with the best validation accuracy.

For DECE, we use $M = 15$ bins and scaling parameters $\tau_a = 100, \tau_b = 0.01$. Learnable label smoothing coefficients are optimised using Adam (Kingma & Ba, 2015) optimiser with learning rate of 0.001. The meta-learnable parameters are initialized at 0.0 (no label smoothing or L2 regularization initially). The total number of meta-parameters is $K \times K$, 1 and $K$ for the non-uniform, scalar and vector label smoothing respectively, while it is $512 \times K + K$ for learnable L2 regularization. We use $\lambda = 0.5$ in the meta-objective, and we have selected it based on validation set calibration and accuracy after trying several values.

**Results** We first follow the experimental setup of Mukhoti et al. (2020) and compare with the following alternatives: 1) cross-entropy, 2) Brier score (Brier, 1950), 3) Weighted MMCE (Kumar et al., 2018) with $\lambda = 2$, 4) Focal loss (Lin et al., 2017) with $\gamma = 3$, 5) Adaptive (sample dependent) focal loss (FLSD) (Mukhoti et al., 2020) with $\gamma = 5$ and $\gamma = 3$ for predicted probability $p \in [0, 0.2)$ and $p \in [0.2, 1)$ respectively. 6) Label smoothing (LS) with a fixed smoothing factor of 0.05. In all cases we report the mean and standard deviation across three repetitions to obtain a more reliable estimate of the performance. In contrast, Mukhoti et al. (2020) report their results on only one run, so in the tables we include our own results for the comparison with the baselines.

We show the test ECE, test ECE after temperature scaling (TS) and test error rates in Tables 1, 2 and 3 respectively. Meta-Calibration leads to excellent intrinsic calibration without the need for post-processing (Table 1), which is practically valuable because post-processing is not always possible (Kim & Yun, 2020) or reliable (Ovadia et al., 2019). However, even after post-processing using TS Meta-Calibration gives competitive performance (Table 2), as evidenced by the best average rank across the considered scenarios. Table 3 shows that Meta-Calibration maintains comparable accuracy to the competitors, even if it does not have the best average rank there. Overall Meta-Calibration leads to significantly better intrinsic calibration, while keeping similar or only marginally worse accuracy.

Table 1: Test ECE (%, ↓): Our Meta-Calibration (MC) leads to excellent intrinsic calibration.

| Dataset | Model | CE | Brier | MMCE | FL-3 | FLSD-53 | LS | MC (Ours) |
|---------|-------|----|-------|------|------|---------|----|-----------|
| CIFAR-10 | ResNet18 | $4.23 \pm 0.15$ | $1.23 \pm 0.03$ | $4.36 \pm 0.16$ | $2.11 \pm 0.09$ | $2.22 \pm 0.04$ | $3.63 \pm 0.06$ | $1.17 \pm 0.26$ |
| | ResNet50 | $4.20 \pm 0.01$ | $1.95 \pm 0.15$ | $4.49 \pm 0.18$ | $1.48 \pm 0.19$ | $1.68 \pm 0.14$ | $2.58 \pm 0.26$ | $1.09 \pm 0.09$ |
| | ResNet110 | $4.81 \pm 0.12$ | $2.58 \pm 0.17$ | $4.20 \pm 0.74$ | $1.82 \pm 0.20$ | $2.16 \pm 0.22$ | $1.96 \pm 0.36$ | $1.07 \pm 0.12$ |
| | WideResNet26-10 | $3.37 \pm 0.11$ | $1.03 \pm 0.08$ | $3.48 \pm 0.06$ | $1.57 \pm 0.32$ | $1.50 \pm 0.15$ | $3.68 \pm 0.10$ | $0.94 \pm 0.10$ |
| CIFAR-100 | ResNet18 | $8.79 \pm 0.59$ | $5.19 \pm 0.18$ | $7.41 \pm 1.30$ | $2.83 \pm 0.27$ | $2.47 \pm 0.12$ | $6.87 \pm 0.29$ | $2.52 \pm 0.35$ |
| | ResNet50 | $12.56 \pm 1.44$ | $4.82 \pm 0.36$ | $9.02 \pm 1.72$ | $4.78 \pm 1.00$ | $5.43 \pm 0.31$ | $5.94 \pm 0.52$ | $3.07 \pm 0.18$ |
| | ResNet110 | $14.96 \pm 0.83$ | $6.52 \pm 0.56$ | $12.29 \pm 1.25$ | $6.64 \pm 1.42$ | $7.38 \pm 0.25$ | $10.69 \pm 0.39$ | $2.80 \pm 0.58$ |
| | WideResNet26-10 | $12.39 \pm 1.44$ | $4.26 \pm 0.30$ | $8.35 \pm 2.79$ | $2.36 \pm 0.13$ | $2.30 \pm 0.36$ | $3.94 \pm 0.96$ | $3.86 \pm 0.34$ |
| SVHN | ResNet18 | $2.98 \pm 0.08$ | $1.94 \pm 0.10$ | $3.14 \pm 0.10$ | $2.69 \pm 0.06$ | $2.83 \pm 0.17$ | $3.88 \pm 0.01$ | $1.14 \pm 0.12$ |
| 20 Newsgroups | Global Pooling CNN | $18.58 \pm 0.80$ | $16.49 \pm 0.70$ | $14.68 \pm 1.03$ | $7.51 \pm 0.51$ | $6.13 \pm 1.84$ | $5.14 \pm 0.64$ | $2.56 \pm 0.38$ |
| | Average rank | 6.4 | 3.5 | 6.1 | 2.8 | 3.1 | 4.8 | 1.3 |

Note that while Brier score, Focal loss and FLSD modify the base model's loss function, our Meta-Calibration corresponds to the vanilla cross-entropy baseline, but where label smoothing is tuned by our DECE-driven hyper-parameter meta-learning rather than being selected by hand.

Table 2: Test ECE with temperature scaling (%, ↓): Our Meta-Calibration (MC) obtains excellent calibration also after temperature scaling.

| Dataset | Model | CE | Brier | MMCE | FL-3 | FLSD-53 | LS | MC (Ours) |
|---|---|---|---|---|---|---|---|---|
| CIFAR-10 | ResNet18 | 1.16 ± 0.10 (2.30) | 1.23 ± 0.03 (1.00) | 1.25 ± 0.08 (2.30) | 1.10 ± 0.11 (0.90) | 1.48 ± 0.31 (0.87) | 1.31 ± 0.08 (0.90) | 1.34 ± 0.47 (0.97) |
| | ResNet50 | 1.20 ± 0.17 (2.53) | 0.97 ± 0.02 (1.17) | 1.35 ± 0.38 (2.50) | 1.10 ± 0.16 (1.07) | 1.21 ± 0.31 (1.07) | 1.42 ± 0.15 (0.90) | 1.09 ± 0.09 (1.00) |
| | ResNet110 | 1.49 ± 0.19 (2.57) | 1.55 ± 0.35 (1.13) | 1.20 ± 0.45 (1.90) | 1.21 ± 0.07 (1.10) | 1.33 ± 0.14 (1.10) | 2.16 ± 0.21 (0.90) | 1.33 ± 0.37 (0.97) |
| | WideResNet26-10 | 1.14 ± 0.13 (2.20) | 1.03 ± 0.08 (1.00) | 0.99 ± 0.19 (2.23) | 1.20 ± 0.29 (0.87) | 1.09 ± 0.02 (0.90) | 1.32 ± 0.04 (0.90) | 0.94 ± 0.10 (1.00) |
| CIFAR-100 | ResNet18 | 5.47 ± 0.22 (1.33) | 4.21 ± 0.23 (0.90) | 6.09 ± 0.39 (1.13) | 2.83 ± 0.27 (1.00) | 2.47 ± 0.12 (1.00) | 4.37 ± 0.45 (0.90) | 2.71 ± 0.58 (1.03) |
| | ResNet50 | 2.51 ± 0.23 (1.57) | 3.43 ± 0.32 (1.10) | 3.19 ± 0.53 (1.37) | 2.25 ± 0.69 (1.10) | 2.53 ± 0.11 (1.10) | 4.28 ± 0.42 (1.10) | 2.51 ± 0.45 (1.07) |
| | ResNet110 | 3.77 ± 0.51 (1.57) | 3.71 ± 0.67 (1.17) | 2.74 ± 0.45 (1.40) | 3.97 ± 0.28 (1.10) | 4.13 ± 0.40 (1.10) | 6.04 ± 0.31 (1.10) | 2.55 ± 0.33 (1.03) |
| | WideResNet26-10 | 3.08 ± 0.26 (1.80) | 2.49 ± 0.13 (1.10) | 4.52 ± 0.52 (1.40) | 2.20 ± 0.12 (1.03) | 2.30 ± 0.36 (1.00) | 3.62 ± 0.74 (1.07) | 2.72 ± 0.19 (1.10) |
| SVHN | ResNet18 | 0.74 ± 0.04 (2.10) | 0.83 ± 0.09 (0.90) | 1.10 ± 0.01 (2.30) | 0.90 ± 0.43 (0.83) | 1.11 ± 0.37 (0.87) | 1.45 ± 0.51 (0.87) | 1.14 ± 0.12 (1.00) |
| 20 Newsgroups | Global Pooling CNN | 2.85 ± 0.34 (3.67) | 4.32 ± 0.79 (2.97) | 4.00 ± 0.22 (2.60) | 3.59 ± 0.34 (1.43) | 2.76 ± 0.20 (1.33) | 3.19 ± 0.30 (1.10) | 2.50 ± 0.32 (0.97) |
| | Average rank | 3.7 | 3.8 | 4.4 | 3.0 | 3.9 | 6.2 | 2.8 |

Table 3: Test error (%, ↓): Our Meta-Calibration (MC) obtains excellent calibration with only small increases in the test error.

| Dataset | Model | CE | Brier | MMCE | FL-3 | FLSD-53 | LS | MC (Ours) |
|---|---|---|---|---|---|---|---|---|
| CIFAR-10 | ResNet18 | 4.99 ± 0.14 | 5.27 ± 0.21 | 5.17 ± 0.19 | 5.06 ± 0.09 | 5.22 ± 0.04 | 4.94 ± 0.13 | 5.22 ± 0.06 |
| | ResNet50 | 4.90 ± 0.02 | 5.15 ± 0.14 | 5.13 ± 0.12 | 5.27 ± 0.22 | 5.26 ± 0.15 | 4.77 ± 0.11 | 5.46 ± 0.05 |
| | ResNet110 | 5.40 ± 0.10 | 5.97 ± 0.17 | 5.70 ± 0.12 | 5.67 ± 0.33 | 5.87 ± 0.13 | 5.45 ± 0.11 | 6.09 ± 0.22 |
| | WideResNet26-10 | 3.99 ± 0.07 | 4.20 ± 0.03 | 4.11 ± 0.06 | 4.18 ± 0.03 | 4.22 ± 0.05 | 4.05 ± 0.07 | 4.36 ± 0.20 |
| CIFAR-100 | ResNet18 | 22.85 ± 0.17 | 23.50 ± 0.17 | 23.80 ± 0.18 | 22.87 ± 0.16 | 23.23 ± 0.32 | 22.35 ± 0.27 | 23.88 ± 0.20 |
| | ResNet50 | 22.41 ± 0.24 | 24.81 ± 0.33 | 22.43 ± 0.05 | 22.27 ± 0.13 | 22.76 ± 0.27 | 21.85 ± 0.06 | 23.22 ± 0.48 |
| | ResNet110 | 22.99 ± 0.19 | 28.29 ± 1.42 | 23.81 ± 0.58 | 23.12 ± 0.26 | 23.71 ± 0.24 | 23.08 ± 0.15 | 24.51 ± 0.41 |
| | WideResNet26-10 | 20.41 ± 0.12 | 20.77 ± 0.05 | 20.60 ± 0.10 | 19.80 ± 0.40 | 19.97 ± 0.25 | 20.82 ± 0.42 | 22.35 ± 0.03 |
| SVHN | ResNet18 | 4.11 ± 0.08 | 3.90 ± 0.19 | 4.15 ± 0.08 | 4.20 ± 0.07 | 4.18 ± 0.06 | 4.13 ± 0.09 | 4.08 ± 0.02 |
| 20 Newsgroups | Global Pooling CNN | 26.64 ± 0.27 | 26.59 ± 0.72 | 26.92 ± 0.32 | 27.65 ± 0.38 | 27.59 ± 0.94 | 26.10 ± 0.31 | 27.26 ± 0.59 |
| | Average rank | 2.1 | 4.9 | 4.2 | 3.9 | 4.8 | 2.1 | 5.9 |

## 4.2 Further analysis

**Alternative calibration metrics** We investigate if models meta-trained using DECE also perform well when evaluated using more advanced calibration metrics than ECE. In particular, we evaluate performance using class-wise ECE (CECE) (Kumar et al., 2019; Widmann et al., 2019; Vaicenavicius et al., 2019; Kull et al., 2019) that considers the scores of all classes in the predicted distribution, instead of only the class with the highest probability. The results in Table 4 confirm that models meta-trained using DECE have excellent calibration also in terms of the CECE criterion.

Table 4: Test Classwise-ECE (%, ↓): Our Meta-Calibration (MC) leads to excellent calibration also when using a more advanced calibration metric.

| Dataset | Model | CE | Brier | MMCE | FL-3 | FLSD-53 | LS-0.05 | MC (Ours) |
|---|---|---|---|---|---|---|---|---|
| CIFAR-10 | ResNet18 | 0.87 ± 0.04 | 0.46 ± 0.02 | 0.91 ± 0.03 | 0.52 ± 0.02 | 0.53 ± 0.03 | 0.73 ± 0.01 | 0.41 ± 0.01 |
| | ResNet50 | 0.88 ± 0.01 | 0.46 ± 0.02 | 0.93 ± 0.04 | 0.44 ± 0.02 | 0.44 ± 0.03 | 0.63 ± 0.02 | 0.48 ± 0.04 |
| | ResNet110 | 0.99 ± 0.02 | 0.58 ± 0.04 | 0.89 ± 0.14 | 0.50 ± 0.02 | 0.55 ± 0.05 | 0.66 ± 0.03 | 0.45 ± 0.01 |
| | WideResNet | 0.71 ± 0.01 | 0.37 ± 0.00 | 0.74 ± 0.01 | 0.43 ± 0.02 | 0.43 ± 0.04 | 0.72 ± 0.02 | 0.34 ± 0.00 |
| CIFAR-100 | ResNet18 | 0.23 ± 0.01 | 0.24 ± 0.00 | 0.22 ± 0.01 | 0.20 ± 0.00 | 0.20 ± 0.00 | 0.26 ± 0.00 | 0.19 ± 0.00 |
| | ResNet50 | 0.29 ± 0.03 | 0.20 ± 0.01 | 0.24 ± 0.03 | 0.20 ± 0.00 | 0.20 ± 0.01 | 0.21 ± 0.00 | 0.19 ± 0.00 |
| | ResNet110 | 0.34 ± 0.02 | 0.23 ± 0.01 | 0.29 ± 0.02 | 0.22 ± 0.01 | 0.23 ± 0.00 | 0.26 ± 0.00 | 0.19 ± 0.00 |
| | WideResNet | 0.29 ± 0.02 | 0.19 ± 0.00 | 0.23 ± 0.03 | 0.18 ± 0.00 | 0.18 ± 0.00 | 0.21 ± 0.01 | 0.19 ± 0.00 |
| SVHN | ResNet18 | 0.62 ± 0.02 | 0.52 ± 0.02 | 0.65 ± 0.02 | 0.67 ± 0.03 | 0.68 ± 0.04 | 0.81 ± 0.05 | 0.29 ± 0.03 |
| 20 Newsgroups | Global Pooling CNN | 2.01 ± 0.07 | 1.80 ± 0.04 | 1.63 ± 0.09 | 1.29 ± 0.03 | 1.22 ± 0.14 | 0.97 ± 0.06 | 1.00 ± 0.04 |
| | Average rank | 6.0 | 3.3 | 5.8 | 2.5 | 2.8 | 5.1 | 1.6 |

**Alternative hyper-parameter choice** We present a general metric that can be used for optimising hyper-parameters for superior calibration. While our main experiments are conducted with non-uniform label smoothing, we demonstrate the generality of the framework by also learning alternative meta-parameters. In

particular, we also consider scalar and vector version of label smoothing as well as learnable L2 regularization. We perform the additional evaluation using ResNet18 on the CIFAR benchmark.

The results in Table 5 confirm learnable L2 regularization also leads to clear improvement in ECE over the cross-entropy baseline. However, the error rate is slightly increased compared to learnable LS, hence we focused on the latter for our other experiments. Scalar and vector LS (MC-S and MC-V) have both improved the calibration, but non-uniform label smoothing (MC) has worked better thanks to its larger expressivity.

Table 5: Comparison of hyper-parameter choice for meta-calibration: CIFAR benchmark with ResNet18 model. Test errors (%, ↓) and test ECE (%, ↓). Other variants of Meta-Calibration also lead to strong improvements in calibration, with non-uniform label smoothing leading to the best calibration overall.

| Dataset | Method | ECE (↓) | Error (↓) |
|---|---|---|---|
| CIFAR-10 | CE | $4.23 \pm 0.15$ | $4.99 \pm 0.14$ |
| | MC | $1.17 \pm 0.26$ | $5.22 \pm 0.06$ |
| | MC-S | $1.48 \pm 0.26$ | $5.17 \pm 0.13$ |
| | MC-V | $1.51 \pm 0.26$ | $5.07 \pm 0.03$ |
| | MC-L2 | $1.78 \pm 0.22$ | $5.49 \pm 0.14$ |
| CIFAR-100 | CE | $8.79 \pm 0.59$ | $22.85 \pm 0.17$ |
| | MC | $2.52 \pm 0.35$ | $23.88 \pm 0.20$ |
| | MC-S | $6.13 \pm 1.20$ | $24.07 \pm 0.17$ |
| | MC-V | $3.98 \pm 0.23$ | $23.96 \pm 0.12$ |
| | MC-L2 | $4.18 \pm 0.26$ | $26.10 \pm 0.14$ |

**Ablation study on meta-learning objective design**   Recall our framework in Equation 1 is setup to perform conventional model training in the inner optimisation, given hyper-parameters; and meta-learning of hyper-parameters in the outer optimisation, by minimising a combination of cross-entropy and our DECE metric as evaluated on the meta-validation set. While we view this setup as being the most intuitive, other architectures are also possible in terms of choice of objective for use in the inner and outer layer of the bilevel optimisation. As a comparison to our DECE, we also evaluate the prior metric MMCE previously proposed as a proxy for model calibration in (Kumar et al., 2018).

From the results in Table 6 we can conclude that: 1) Meta-learning with combined CE and DECE meta-objective is beneficial for improving calibration (M5 vs M0). 2) Alternative outer-loop objectives CE (M2) and DECE (M3) improve calibration but not as significantly as the combined meta-objective (M5 vs M2 and M5 vs M4). 3) MMCE completely fails as a meta-objective (M3). 4) DECE improves calibration when used as a secondary loss in multi-task learning, but at greater detriment to test error (M1 vs M0). 5) Our combined meta-objective (M5) is the best overall.

Table 6: Ablation study on losses for inner and outer objectives in bilevel optimisation using CIFAR-10 and CIFAR-100 with ResNet18.

| Model | Meta-Loss | Loss | CIFAR-10 | | CIFAR-100 | |
|---|---|---|---|---|---|---|
| | | | ECE (%, ↓) | Error (%, ↓) | ECE (%, ↓) | Error (%, ↓) |
| M0: Vanilla CE | - | CE | $4.23 \pm 0.15$ | $4.99 \pm 0.14$ | $8.79 \pm 0.59$ | $22.85 \pm 0.17$ |
| M1: Multi-task | - | CE + DECE | $3.80 \pm 0.03$ | $10.24 \pm 0.21$ | $4.40 \pm 0.39$ | $29.49 \pm 0.17$ |
| M2: Meta-Calibration | CE | CE | $1.31 \pm 0.36$ | $5.13 \pm 0.24$ | $3.00 \pm 1.12$ | $23.72 \pm 0.40$ |
| M3: Meta-Calibration | MMCE | CE | $44.24 \pm 0.70$ | $6.77 \pm 0.25$ | $21.94 \pm 2.39$ | $25.40 \pm 0.31$ |
| M4: Meta-Calibration | DECE | CE | $1.26 \pm 0.44$ | $5.21 \pm 0.14$ | $3.28 \pm 0.31$ | $23.83 \pm 0.14$ |
| M5: Meta-Calibration | CE + DECE | CE | $1.17 \pm 0.26$ | $5.22 \pm 0.06$ | $2.52 \pm 0.35$ | $23.88 \pm 0.20$ |

**Evaluating DECE approximation to ECE**   A key contribution of this work is DECE, a differentiable approximation to expected calibration error. In this section we investigate the quality of our DECE approximation. We trained the same ResNet18 backbone on both CIFAR-10 and CIFAR-100 benchmarks

for 350 epochs, recording DECE and ECE values at various points. The results in Figure 2a show both Spearman and Pearson correlation coefficient between DECE and ECE. In both cases they are close to 1, and become even closer to 1 as training continues. This shows that DECE accurately estimates ECE, while providing differentiability for end-to-end optimisation. We further we show in Figure 2b that their mean values are very close to each other.

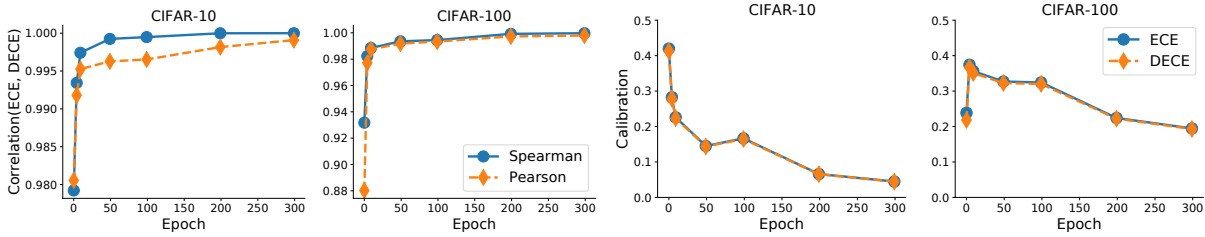

(a) Correlation between DECE and ECE is close to 1.     (b) Mean ECE and DECE are close to each other.

Figure 2: Evaluation of how DECE approximates ECE, using ResNet18 on CIFAR.

**What hyper-parameters are learned?** We show our approach learns non-trivial hyper-parameter settings to achieve its excellent calibration performance. Figure 3 shows how the learned overall strength of smoothing evolves during training for both CIFAR-10 and CIFAR-100 benchmarks – using ResNet18. We show the mean and standard deviation across three repetitions and all classes.

From the figure we observe label smoothing changes in response to changes in learning rate, which happens after 150 and 250 epochs. For CIFAR-100 with more classes it starts with large smoothing values and finishes with smaller values. The large standard deviations are due to the model making use of a wide range of class-wise smoothing parameters. It would be infeasible to manually select a curriculum for label smoothing at different stages of training, as it would be to tune a range of smoothing parameters: The ability to optimise these hyper-parameters automatically is a key benefit of our framework.

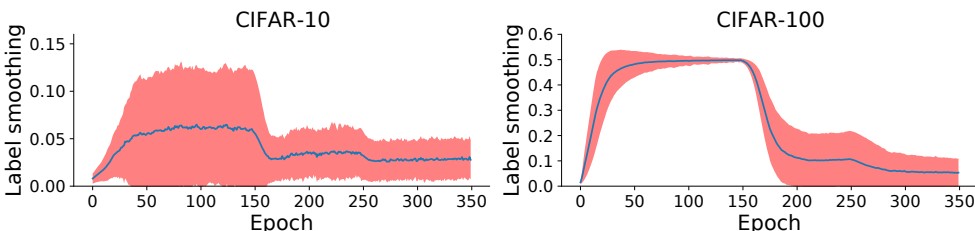

Figure 3: Overall label smoothing during training for CIFAR, using ResNet18. The learned smoothing strategy is non-trivial and adapts according to the learning rate schedule.

We also analyse how the smoothing is distributed across the different classes in Figure 4 and 5. The results show that the smoothing is indeed non-uniform, demonstrating the model does exploit the ability to learn a complex label-smoothing distribution. The learned non-uniform label-smoothing distribution can be observed to subject visually similar classes to more smoothing (Figure 4(b)), which makes sense to reduce the confidence of the most likely kinds of specific errors. This idea is quantified more systematically for CIFAR-100 in Figure 5, which compares the average degree of smoothing between classes in the same superclass, and those in different superclasses. The results show that within-superclass smoothing is generally much stronger than across-superclass smoothing, even though the model receives no annotation or supervision about superclasses. It learns this smoothing structure given the objective of optimising (meta-)validation calibration.

We further analysed the hyper-parameters in the case of learnable L2 regularization and show it as part of the appendix. The figure shows we learn a range of regularization values to achieve a good calibration outcome.

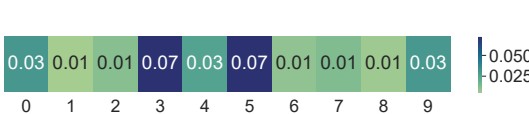
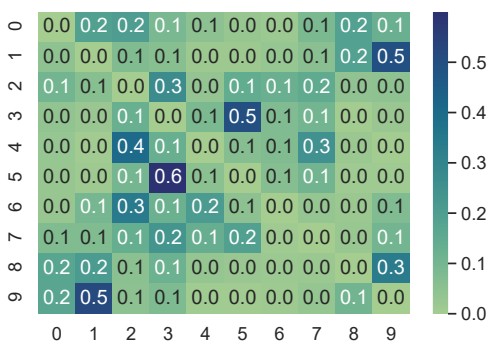

(a) Overall level of smoothing in various classes. (b) Distribution of smoothing across various classes.

Figure 4: Analysis of learned non-uniform label smoothing for CIFAR-10, using ResNet18 model. Visually similar classes receive more smoothing – e.g. cat and dog (3 and 5), and automobile and truck (1 and 9).

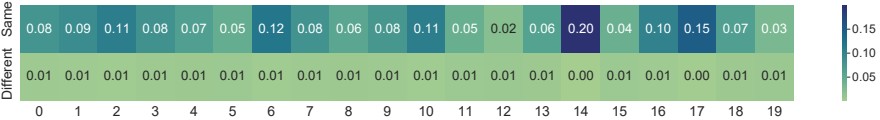

Figure 5: Meta-Calibration learns to give more label smoothing within the same superclass compared to other superclasses in CIFAR-100, using ResNet18 model.

This highlights the value of our differentiable framework that enables efficient gradient-based optimisation of many hyper-parameters.

**Analysis of accuracy vs calibration using simulated data** Meta-Calibration leads to significantly better calibration, while keeping similar or only marginally worse accuracy. To study if there is a trade-off between accuracy vs calibration we perform an experiment using simulated data. More specifically the experiment with simulated data consists of 1) sampling parameters of a binary logistic regression model (oracle) over 2D data, 2) sampling the label for each data point from a binomial distribution with a class probability given by the oracle. 3) We then use the sampled data to train (i) a vanilla logistic regression model, (ii) label-smoothing logistic regression model and (iii) a Meta-Calibration logistic regression model. This enables us to compare the best-case classifier accuracy and calibration with the results of learned models. Our experiment is repeated across three random seeds so that we can report the mean and standard deviation.

The results in Table 7 confirm Meta-Calibration matches both the test accuracy and ECE of the oracle, obtaining close to the best possible calibration. Analogous cross-entropy training as well as simple label smoothing (LS) have a significantly larger ECE. The ECE of Meta-Calibration and oracle is close to 0, but not precisely 0 due to sampling effects (i.e. because we do not use an infinite amount of data). We have also visualized the estimated class probabilities of different data points, together with the decision boundary in Figure 6. The visualization shows that Meta-Calibration and the oracle are similarly calibrated (e.g., similar point shades close to the decision boundary), while a difference in calibration (point shading) is perceptible for cross-entropy and LS.

## 5 Discussion

This work is a pioneering step in using meta-learning to directly optimise model calibration. Learnable rather than hand-tuned calibration is important as different models and datasets have very different calibration properties, precluding a one-size-fits-all solution (Minderer et al., 2021). There are many ways our work could

Table 7: Analysis of accuracy and calibration on simulated data with known oracle. Test accuracy and ECE (%) are reported, with the mean and standard deviation computed across three random seeds. Our Meta-Calibration (MC) matches both the accuracy and ECE of the oracle, obtaining close to perfect calibration.

| Metric | Oracle | Cross-Entropy | LS | MC (Ours) |
|---|---|---|---|---|
| Accuracy ($\uparrow$) | $87.62 \pm 1.87$ | $87.55 \pm 1.86$ | $87.57 \pm 1.87$ | $87.55 \pm 1.81$ |
| ECE ($\downarrow$) | $1.38 \pm 0.32$ | $9.81 \pm 1.14$ | $3.61 \pm 0.43$ | $1.40 \pm 0.19$ |

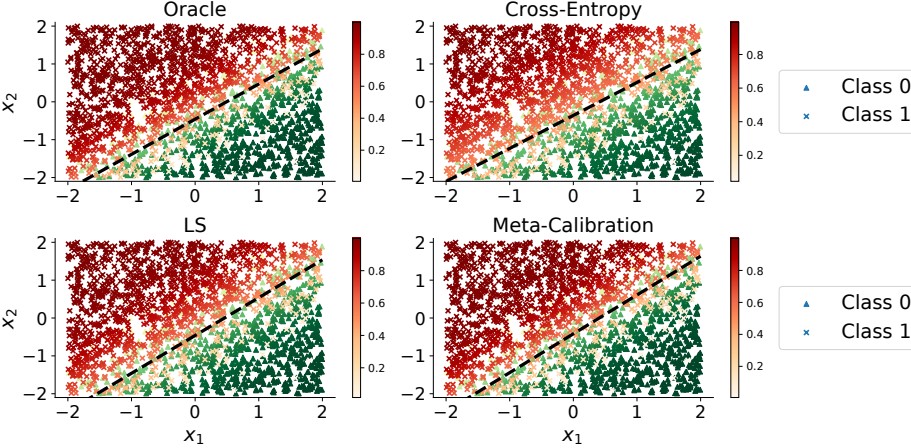

Figure 6: Visualization of the estimated probabilities across test data points, with the decision boundary.

be extended in the future. One direction is to target different hyper-parameters beyond the label smoothing and L2 classifier regularization evaluated here, such as loss learning. While we did not explore this here, the framework could also be used to unify post-hoc methods such as temperature scaling by treating temperature as the hyper-parameter. Secondly, better meta-learning algorithms such as implicit meta-learning (Lorraine et al., 2020) could better optimise the meta-objective. A third direction is to extend the differentiable metric itself to e.g. adapt it to various domain-specific calibration measures – for example ones relevant to the finance industry (Liu et al., 2019).

A drawback of our approach is the computational overhead added by meta-learning compared to basic model training. However, it is still manageable and may be worth it when well-calibrated models are crucial. We give an overview in Table 8 in the appendix. Ongoing advances in efficient meta-learners (Bohdal et al., 2021) can make the overhead smaller. In terms of social implications, our work aims to improve the reliability of neural networks, but there still are risks the neural networks will fail to accurately estimate their confidence.

## 6 Conclusion

We introduced a new DECE metric that accurately represents the common calibration ECE measure and makes it differentiable. With DECE, we can directly optimise hyper-parameters for calibration and obtain competitive results with hand-designed architectures. We believe DECE opens up a new avenue for the community to tackle the challenge of model calibration in optimisation-based ways.

**Acknowledgments**

This work was supported in part by the EPSRC Centre for Doctoral Training in Data Science, funded by the UK Engineering and Physical Sciences Research Council (grant EP/L016427/1) and the University of Edinburgh.

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

## A   Learned L2 regularization values

We show the learned unit-wise L2 classifier regularization values in Figure 7. The results show there is a large variability in the coefficients and they take both positive and negative values.

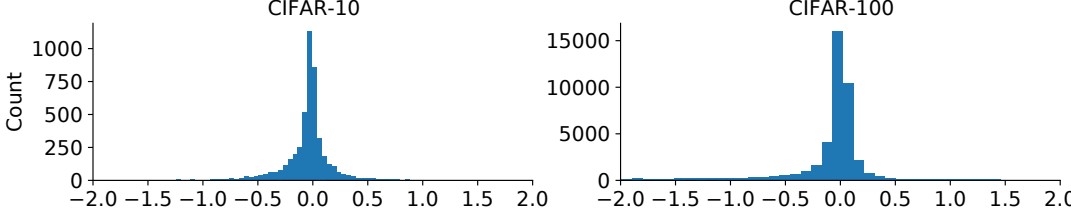

Figure 7: Histograms of the learned unit-wise L2 classifier regularization values.

## B   Reliability analysis

We perform reliability analysis and show the percentage of samples with various confidence levels in Figure 8 for CIFAR-10 and CIFAR-100. We use ResNet18 and take the best model from training – early stopping. The figure shows learnable label smoothing leads to visually better alignment between the expected and actual confidence binning. It also leads to softening the confidences of predictions, which is expected for label smoothing.

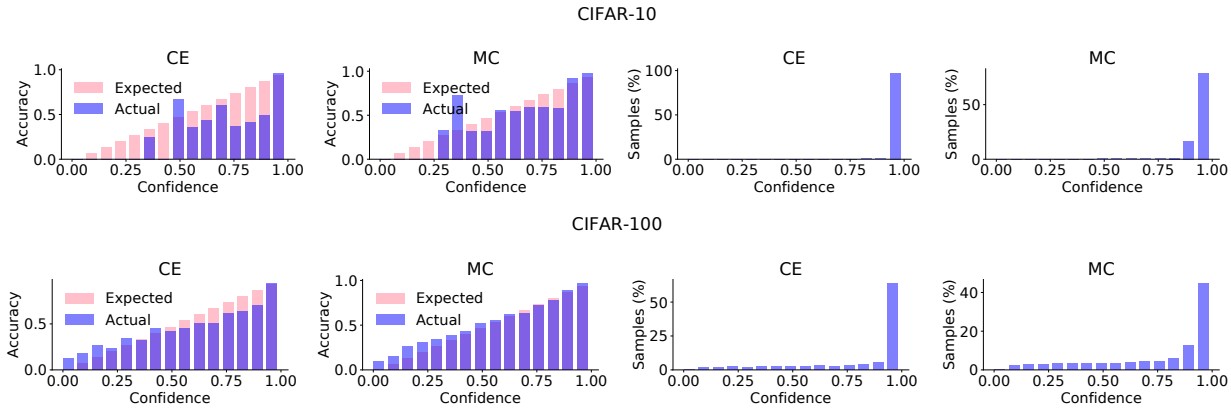

Figure 8: Reliability analysis for CIFAR-10 and CIFAR-100, using ResNet18 model.

## C   Training times analysis

We report the training times of the different approaches in Table 8 (for simplicity we only include CE baseline as all baselines have similar training time). We see that even if training with Meta-Calibration takes longer, the overall training time remains manageable. If excellent calibration is key, longer training time is acceptable.

Table 8: Training times in hours. We used one NVIDIA Titan X for each experiment. Meta-Calibration takes longer, but if excellent calibration is a priority, the additional time is acceptable.

| Dataset | Model | CE | MC (Ours) |
|---|---|---|---|
| CIFAR-10 | ResNet18 | 2.8h ± 0.1h | 7.0h ± 0.2h |
| | ResNet50 | 8.9h ± 0.1h | 21.9h ± 0.1h |
| | ResNet110 | 17.3h ± 0.3h | 44.6h ± 7.1h |
| | WideResNet26-10 | 9.2h ± 0.3h | 29.7h ± 0.2h |
| CIFAR-100 | ResNet18 | 2.7h ± 0.1h | 12.4h ± 0.2h |
| | ResNet50 | 8.9h ± 0.1h | 26.4h ± 0.2h |
| | ResNet110 | 17.3h ± 0.2h | 50.5h ± 1.0h |
| | WideResNet26-10 | 9.3h ± 0.4h | 39.4h ± 6.7h |
| SVHN | ResNet18 | 4.4h ± 0.6h | 9.6h ± 0.1h |
| 20 Newsgroups | Global Pooling CNN | 0.1h ± 0.0h | 0.3h ± 0.1h |

## D   Cross-domain evaluation

**Setup**   To evaluate our framework's calibration performance under distribution shift, we use the CIFAR-C benchmark (Hendrycks & Dietterich, 2019). CIFAR-C contains a relatively large variety of image corruptions, such as adding fog, pixelation, changes to the brightness and many others (overall 19 corruption types). There are 5 levels of severity for each of them, which can be interpreted as providing 95 domains. We select 22 of these as test domains following (Zhang et al., 2021). These include impulse noise, motion blur, fog, and elastic transform at all levels of severity, and spatter and JPEG compression at the maximum level of severity. All other domains from (Hendrycks & Dietterich, 2019) are used during training and validation.

All approaches are trained with augmented (corrupted) data so that they can better generalize across domains. In each step we sample a corruption to use from the set of training corruptions. Meta-Calibration in particular samples a separate corruption for the inner and outer loop so that it can train meta-parameters that are more likely to generalize to new domains.

**Results**   We have evaluated our approach using CIFAR-C benchmark and ResNet18 model, and we show the results in Table 9 and 10. We include both the average across all test domains as well as the result on the most challenging domain (worst case). We report the mean and standard deviation across three repetitions. Meta-Calibration is still helpful for obtaining better calibration in the cross-domain case, but not to as large an extent as for clean data. While the efficacy of this algorithm likely depends on whether the domain-shifts seen during meta-training are representative in strength of those seen during meta-testing, we still view these as encouraging initial results that tools from multi-domain meta-learning can be adapted to address model calibration under domain shift. Studying how to more successfully exploit meta-learning for calibration in cross-domain scenarios is an interesting research question that the ML community can focus on in the future.

Table 9: Test ECE (%, ↓) for our cross-domain experiments on CIFAR-C.

| Dataset | Case | CE | LS | Brier | MMCE | FL | FLSD | MC (Ours) |
|---|---|---|---|---|---|---|---|---|
| CIFAR-10-C | Average | 7.28 ± 0.03 | 3.13 ± 0.19 | 4.20 ± 0.81 | 4.12 ± 0.23 | 4.02 ± 0.26 | 4.13 ± 0.01 | 3.00 ± 0.28 |
| | Worst case | 15.55 ± 0.32 | 9.43 ± 0.65 | 9.19 ± 1.84 | 5.47 ± 0.78 | 4.97 ± 0.64 | 9.05 ± 0.88 | 10.97 ± 0.99 |
| CIFAR-100-C | Average | 7.77 ± 0.09 | 5.13 ± 0.05 | 4.89 ± 0.06 | 6.07 ± 0.59 | 6.33 ± 0.23 | 7.06 ± 0.54 | 5.26 ± 2.03 |
| | Worst case | 14.75 ± 1.09 | 9.42 ± 0.71 | 7.40 ± 0.32 | 8.30 ± 0.75 | 8.53 ± 0.16 | 8.94 ± 0.61 | 6.79 ± 2.37 |

Table 10: Test error (%, ↓) for our cross-domain experiments on CIFAR-C.

| Dataset | Case | CE | LS | Brier | MMCE | FL | FLSD | MC (Ours) |
|---|---|---|---|---|---|---|---|---|
| CIFAR-10-C | Average | $10.60 \pm 0.05$ | $10.55 \pm 0.19$ | $10.67 \pm 0.11$ | $11.08 \pm 0.18$ | $11.05 \pm 0.15$ | $10.13 \pm 0.04$ | $11.22 \pm 0.22$ |
| | Worst case | $21.83 \pm 0.49$ | $21.78 \pm 0.18$ | $21.32 \pm 0.42$ | $23.52 \pm 0.29$ | $23.34 \pm 0.79$ | $21.03 \pm 0.80$ | $23.57 \pm 0.78$ |
| CIFAR-100-C | Average | $34.20 \pm 0.09$ | $35.26 \pm 0.09$ | $34.27 \pm 0.23$ | $34.26 \pm 0.22$ | $34.14 \pm 0.38$ | $33.32 \pm 0.10$ | $34.68 \pm 0.13$ |
| | Worst case | $57.10 \pm 0.98$ | $56.56 \pm 0.91$ | $56.95 \pm 0.27$ | $56.00 \pm 0.20$ | $55.87 \pm 1.47$ | $54.69 \pm 0.44$ | $55.76 \pm 0.51$ |

# E   Comparison to SB-ECE

Soft-binned ECE (SB-ECE) approximates the binning operation in ECE so that it is differentiable and can be used as an auxiliary loss during training (Karandikar et al., 2021). Comparing DECE with SB-ECE, our DECE has both conceptual and empirical advantages. Conceptual advantages are as follows: 1) We also make the accuracy component of ECE differentiable, 2) SB-ECE binning estimate for the left-most and right-most bin can be inaccurate as a result of using bin's center value, while our binning approach does not suffer from this. The empirical advantages are:

- DECE provides a closer approximation to ECE than SB-ECE as empirically evaluated in Figure 9 and 10. The quality of binning in SB-ECE is controlled by temperature parameter $T$, and we try both lower $T = 0.0001$ and higher temperatures $T = 0.01$. The results show DECE provides a significantly better approximation to ECE than SB-ECE regardless the value of the temperature. In fact, we see that the quality of SB-ECE approximation is relatively insensitive to the value of the temperature parameter $T$.

- Our Meta-Calibration with DECE leads to better calibration. Karandikar et al. (2021) propose SB-ECE and SAvUC (soft version of accuracy versus uncertainty calibration loss (Krishnan & Tickoo, 2020)) to be used as auxiliary losses to encourage better calibration. We evaluate SB-ECE and SAvUC on our benchmark setup, both in their original form (as a regularizer added to CE or Focal Loss) and in our meta-learning framework (as part of meta-objective for LS learning) in Table 11. The results confirm the benefits of using Meta-Calibration with meta-objective that includes DECE. We additionally include comparison of error rates in Table 12. The error rates remain similar to Meta-Calibration and the other baselines, although instabilities can occur that lead to noticeably worse error rates.

Karandikar et al. (2021) also introduce an alternative way of training called interleaved training. In such training each epoch is split into two and a separate set is used for training with respect to calibration. We have implemented and evaluated interleaved training as described in (Karandikar et al., 2021), and we report test ECE and error (%) in Table 13 and 14 respectively. The results suggest interleaved training generally leads to worse ECE than our approach, in most cases significantly worse. Interleaved training also leads to significantly worse error compared to our Meta-Calibration and the other baselines. We attribute the empirical benefits of Meta-Calibration to using the calibration objective for training only the label smoothing meta-parameters and also the specialised meta-training that uses an inner and outer loop.

Table 11: Test ECE (%, ↓) – comparison of Meta-Calibration (MC) that uses DECE vs SB-ECE and SAvUC.

| Dataset | Model | CE+SBECE | CE+SAvUC | FL3+SBECE | FL3+SAvUC | MC-SBECE | MC-SAvUC | MC (Ours) |
|---|---|---|---|---|---|---|---|---|
| CIFAR-10 | ResNet18 | $5.37 \pm 0.16$ | $3.26 \pm 0.02$ | $1.31 \pm 0.01$ | $1.91 \pm 0.09$ | $2.63 \pm 0.66$ | $4.18 \pm 0.49$ | $1.17 \pm 0.26$ |
| | ResNet50 | $3.29 \pm 0.37$ | $3.36 \pm 0.10$ | $1.85 \pm 0.03$ | $1.54 \pm 0.17$ | $2.51 \pm 0.90$ | $2.50 \pm 0.42$ | $1.09 \pm 0.09$ |
| CIFAR-100 | ResNet18 | $5.21 \pm 1.15$ | $5.68 \pm 0.26$ | $2.77 \pm 0.08$ | $2.58 \pm 0.10$ | $5.72 \pm 0.21$ | $5.49 \pm 0.54$ | $2.52 \pm 0.35$ |
| | ResNet50 | $5.72 \pm 0.39$ | $12.34 \pm 0.09$ | $5.86 \pm 0.01$ | $4.93 \pm 0.18$ | $3.10 \pm 0.13$ | $7.53 \pm 0.63$ | $3.07 \pm 0.18$ |

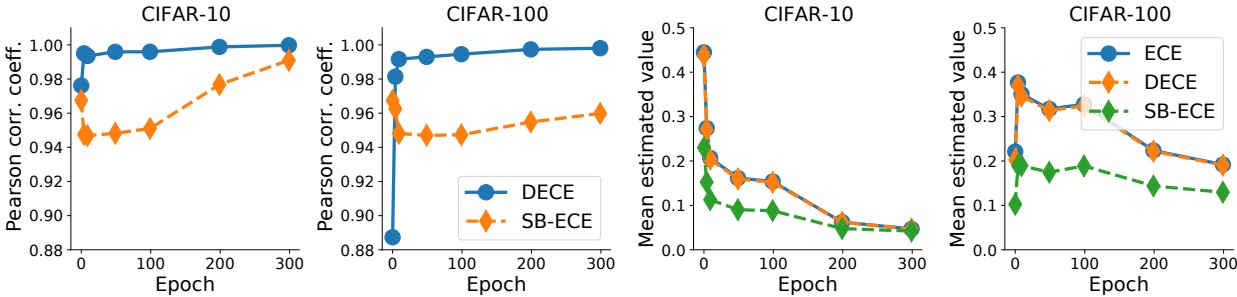

Figure 9: $T = 0.01$: Pearson correlation coefficient between ECE/DECE and ECE/SB-ECE, and the mean estimated value of ECE, DECE and SB-ECE for CIFAR-10 and CIFAR-100 using ResNet18.

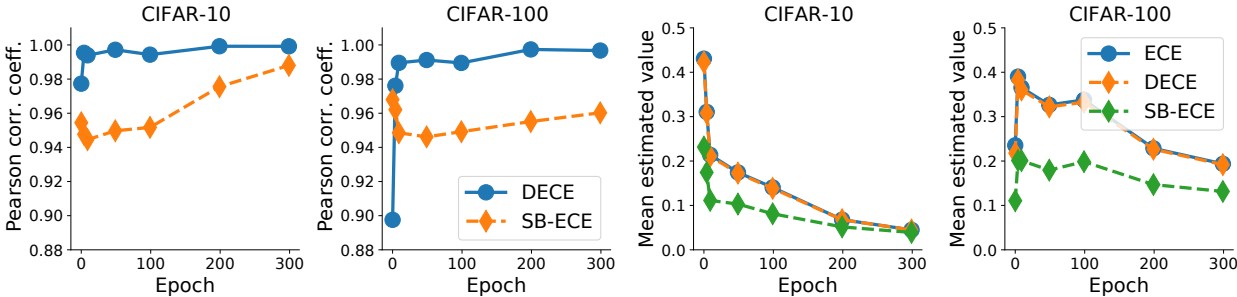

Figure 10: $T = 0.0001$: Pearson correlation coefficient between ECE/DECE and ECE/SB-ECE, and the mean estimated value of ECE, DECE and SB-ECE for CIFAR-10 and CIFAR-100 using ResNet18.

Table 12: Test error (%, ↓) – comparison of Meta-Calibration (MC) that uses DECE vs SB-ECE and SAvUC.

| Dataset | Model | CE+SBECE | CE+SAvUC | FL3+SBECE | FL3+SAvUC | MC-SBECE | MC-SAvUC | MC (Ours) |
|---|---|---|---|---|---|---|---|---|
| CIFAR-10 | ResNet18 | $8.97 \pm 0.04$ | $5.07 \pm 0.06$ | $5.16 \pm 0.10$ | $5.20 \pm 0.11$ | $5.10 \pm 0.12$ | $5.17 \pm 0.25$ | $5.22 \pm 0.06$ |
| | ResNet50 | $12.39 \pm 0.39$ | $5.03 \pm 0.10$ | $5.04 \pm 0.10$ | $5.26 \pm 0.23$ | $5.30 \pm 0.05$ | $5.16 \pm 0.21$ | $5.46 \pm 0.05$ |
| CIFAR-100 | ResNet18 | $27.55 \pm 0.18$ | $22.64 \pm 0.32$ | $22.87 \pm 0.34$ | $22.88 \pm 0.26$ | $23.98 \pm 0.14$ | $23.83 \pm 0.25$ | $23.88 \pm 0.20$ |
| | ResNet50 | $28.33 \pm 0.39$ | $22.09 \pm 0.13$ | $22.40 \pm 0.21$ | $22.02 \pm 0.41$ | $23.51 \pm 0.45$ | $23.18 \pm 0.26$ | $23.22 \pm 0.48$ |

Table 13: Test ECE (%, ↓) – comparison of interleaved training and Meta-Calibration (MC).

| Dataset | Model | CE+SBECE | CE+SAvUC | FL3+SBECE | FL3+SAvUC | MC (Ours) |
|---|---|---|---|---|---|---|
| CIFAR-10 | ResNet18 | $4.81 \pm 1.25$ | $4.67 \pm 0.12$ | $6.90 \pm 0.18$ | $1.52 \pm 0.08$ | $1.17 \pm 0.26$ |
| | ResNet50 | $3.44 \pm 0.37$ | $4.55 \pm 0.15$ | $6.38 \pm 0.82$ | $1.38 \pm 0.31$ | $1.09 \pm 0.09$ |
| CIFAR-100 | ResNet18 | $5.17 \pm 1.23$ | $10.04 \pm 0.65$ | $4.81 \pm 1.05$ | $1.73 \pm 0.24$ | $2.52 \pm 0.35$ |
| | ResNet50 | $6.74 \pm 0.29$ | $11.56 \pm 2.98$ | $4.35 \pm 0.48$ | $2.86 \pm 0.27$ | $3.07 \pm 0.18$ |

Table 14: Test error (%, ↓) – comparison of interleaved training and Meta-Calibration (MC).

| Dataset | Model | CE+SBECE | CE+SAvUC | FL3+SBECE | FL3+SAvUC | MC (Ours) |
|---|---|---|---|---|---|---|
| CIFAR-10 | ResNet18 | $19.09 \pm 5.56$ | $5.66 \pm 0.13$ | $14.22 \pm 0.63$ | $5.96 \pm 0.13$ | $5.22 \pm 0.06$ |
| | ResNet50 | $13.51 \pm 1.01$ | $5.56 \pm 0.13$ | $12.63 \pm 0.56$ | $6.15 \pm 0.45$ | $5.46 \pm 0.05$ |
| CIFAR-100 | ResNet18 | $37.66 \pm 2.85$ | $26.94 \pm 1.62$ | $39.97 \pm 5.71$ | $26.64 \pm 0.47$ | $23.88 \pm 0.20$ |
| | ResNet50 | $35.18 \pm 1.24$ | $24.75 \pm 0.39$ | $35.61 \pm 2.22$ | $25.45 \pm 0.97$ | $23.22 \pm 0.48$ |

## F   Comparison with the setup of Guo et al. (2017)

Our main results adopt the setup from Mukhoti et al. (2020) as they provide a complete codebase and full details of their experiments. In this section we compare this setup with the one in (Guo et al., 2017) that reports better ECE values in their paper. The comparison will focus on ResNet110 model trained with cross-entropy as it is the only model present in both our results and the results from Guo et al. (2017).

We have tried to estimate the details of the training used by Guo et al. (2017), based on the relatively limited information in the paper and the parameters used in the demo code for a different model. After estimating the hyper-parameters of (Guo et al., 2017), the differences compared to us are the following: Guo et al. (2017) train for 500 epochs and decay the learning rate by a factor of 10 after 250 and 375 epochs. In our main experiments we train for 350 epochs and decay the learning rate after 150 and 250 epochs. Minibatch size is 64 in (Guo et al., 2017), while we use 80 for ResNet110 (in general we use 128, but it was downscaled to 80 to fit into memory of our GPUs). We use weight decay of 5e-4 and no Nesterov momentum, while Guo et al. (2017) use Nesterov momentum and potentially use weight decay of 0.001. The demo code provides the value of weight decay, but it is then not used, so we have tried both including and excluding it. While the demo code uses 300 epochs, the paper says that 500 epochs are used for ResNet110, so we have used 500 epochs for the additional experiments. Further, the way that temperature scaling (TS) is done in (Mukhoti et al., 2020) and (Guo et al., 2017) is different. Mukhoti et al. (2020) use grid search over a range of values and argue that this gives stronger baselines than selecting the temperature by minimising validation set NLL. While we follow Mukhoti et al. (2020) in our paper, we also evaluate the alternative option. Based on the demo code, we try 50 iterations for optimising the temperature, but we also try 10 iterations because that value is mentioned in the paper of Guo et al. (2017).

We report the results of our additional experiments in Table 15. The results from the respective papers are at the top of each part, from which we observe that the stochastic depth model is better than the standard model. However, our comparison will focus on the standard model. Comparing (Guo et al., 2017) and (Mukhoti et al., 2020) we see that the test error in (Guo et al., 2017) is significantly worse, as a result of the differences in the training and evaluation procedure. This is likely to relate to the general observation in (Guo et al., 2017) that "the network learns better classification accuracy at the expense of well-modelled probabilities". The two papers report results from one run only, but for our experiments we report mean and standard deviation across three runs.

Our experiments that try to match the setup of Guo et al. (2017) show that grid-search based temperature scaling from (Mukhoti et al., 2020) is significantly better, confirming the observations of Mukhoti et al. (2020). We also observe that weight decay is beneficial, but while it gets us closer to the reported calibration, there is a significant difference in the error rate. Overall we see that neither of the configurations obtains similar values to the ones reported in (Guo et al., 2017) for both CIFAR-10 and CIFAR-100 datasets and across all metrics. One of the configurations that we have evaluated "With weight decay, grid search TS" slightly improves over the setup from Mukhoti et al. (2020), but it comes at an increased runtime cost (500 epochs instead of 350). To summarize, our additional comparison with the setup of Guo et al. (2017) has provided a more detailed justification for following the setup of Mukhoti et al. (2020) in our paper.

Table 15: Comparison of different setups for ResNet110 model. The results suggest neither configuration obtains similar results as reported in (Guo et al., 2017) for both CIFAR-10 and CIFAR-100 datasets.

| Dataset | Setup | Test ECE (%, ↓) | Test ECE with TS (%, ↓) | Test error (%, ↓) |
|---|---|---|---|---|
| CIFAR-10 | (Guo et al., 2017) standard model | 4.60 | 0.83 | 6.21 |
| | (Guo et al., 2017) stochastic depth model | 4.12 | 0.60 | 5.64 |
| | (Mukhoti et al., 2020) standard model | 4.41 | 1.09 | 4.89 |
| | No weight decay, grid-search TS | $5.96 \pm 0.04$ | $1.50 \pm 0.24$ | $6.39 \pm 0.07$ |
| | No weight decay, optim. TS (50 iters) | $5.53 \pm 0.12$ | $5.16 \pm 0.11$ | $5.93 \pm 0.14$ |
| | No weight decay, optim. TS (10 iters) | $5.53 \pm 0.12$ | $5.28 \pm 0.12$ | $5.93 \pm 0.14$ |
| | With weight decay, grid-search TS | $4.04 \pm 0.20$ | $0.75 \pm 0.13$ | $4.56 \pm 0.29$ |
| | With weight decay, optim. TS (50 iters) | $4.13 \pm 0.12$ | $3.59 \pm 0.18$ | $4.63 \pm 0.06$ |
| | With weight decay, optim. TS (10 iters) | $4.13 \pm 0.12$ | $3.80 \pm 0.16$ | $4.63 \pm 0.06$ |
| | Our cross-entropy | $4.81 \pm 0.12$ | $1.49 \pm 0.19$ | $5.40 \pm 0.10$ |
| | Our Meta-Calibration | $1.07 \pm 0.12$ | $1.33 \pm 0.37$ | $6.09 \pm 0.22$ |
| CIFAR-100 | (Guo et al., 2017) standard model | 16.53 | 1.26 | 27.83 |
| | (Guo et al., 2017) stochastic depth model | 12.67 | 0.96 | 24.91 |
| | (Mukhoti et al., 2020) standard model | 19.05 | 4.43 | 22.73 |
| | No weight decay, grid-search TS | $25.68 \pm 0.50$ | $1.85 \pm 0.33$ | $28.71 \pm 0.55$ |
| | No weight decay, optim. TS (50 iters) | $25.88 \pm 0.27$ | $22.84 \pm 0.22$ | $29.24 \pm 0.27$ |
| | No weight decay, optim. TS (10 iters) | $25.88 \pm 0.27$ | $23.96 \pm 0.23$ | $29.24 \pm 0.27$ |
| | With weight decay, grid-search TS | $16.28 \pm 0.22$ | $3.54 \pm 0.28$ | $21.82 \pm 0.19$ |
| | With weight decay, optim. TS (50 iters) | $16.73 \pm 0.07$ | $11.02 \pm 0.20$ | $22.17 \pm 0.24$ |
| | With weight decay, optim. TS (10 iters) | $16.73 \pm 0.07$ | $12.86 \pm 0.14$ | $22.17 \pm 0.24$ |
| | Our cross-entropy | $14.96 \pm 0.83$ | $3.77 \pm 0.51$ | $22.99 \pm 0.19$ |
| | Our Meta-Calibration | $2.80 \pm 0.58$ | $2.55 \pm 0.33$ | $24.51 \pm 0.41$ |

