# OpenReview forum: "Meta-Calibration: Learning of Model Calibration Using Differentiable Expected Calibration Error"
_TMLR — Accepted by TMLR_

### Review · Reviewer_4bkh · 2023-05-28

**Summary Of Contributions:**

This paper offers a differential ECE objective to be minimized during training, on a disjoint subset of “validation” data. One interesting feature of this work is the use of meta-learning combined with label smoothing to improve the model's overall calibration. Overall, the classifiers fitted with this method enjoy impressive ECE although the accuracy is lower than that of standard classifiers trained via the cross entropy loss.

*First contribution: Differentiable ECE*

The authors suggest using an all-pairs approach (Qin et al., 2010) to approximate the accuracy and soft binning (Yang et al., 2018) to approximate the overall ECE metric.

*Second contribution: Meta-learning*

After defining the meta-learning objective (both for L2 regularization on the classifier’s weights and label smoothing), the authors found that optimizing the hyper-parameters of label smoothing leads to better performance. This step's goal, inspired by previous work by Müller et al. (2019), is to learn one or more coefficients to smooth the one-hot encoded labels.

Note that both contributions are only of marginal novelty, as these ideas were suggested in the previous work.

**Audience:**

Yes

**Broader Impact Concerns:**

None.

**Claims And Evidence:**

Yes

**Requested Changes:**

See the weakness above.

**Strengths And Weaknesses:**

**Strengths**

1. Impressive calibration error on the studied data sets.
2. The paper is clear and well-written.

**Weaknesses**

Beyond the issue of marginal novelty compared to previous work, here are concrete concerns that I have regarding the proposed method.

1. *Comparisons with baseline methods.* In particular, the method by Karandikar et al. (2021) seems to be very relevant, as they also offered a differentiable soft-binned ECE loss to improve the calibration metric.  In the paper, the comparison between the methods is deferred to the appendix, which is okay. However, the authors did not include a comparison between the accuracy of the two methods. Thus it is not clear whether the method proposed in this work is superior both in terms of calibration and accuracy. The second method that is of interest to compare is the one by Müller et al. (2019(, which also combines meta-learning with label smoothing. This comparison is meaningful as label smoothing has been shown to improve the ECE calibration metric.

2. *Accuracy of the classifier obtained by the proposed method compared to other techniques.* I find it disappointing that, for example, the accuracy of FLSD-53 with wide resnet is better than the proposed method while having a comparable ECE.

3. This brings me to the third concern: it is *not clear how to handle the trade-off in accuracy vs. calibration*. I would love to see an experiment with *simulated* data where an oracle classifier is known – i.e., the classifier has access to the true conditional distribution of Y|X, from which the label is sampled. Importantly, the oracle in such an experiment is calibrated since we have access to the true conditional class probabilities. With this at hand, it will be important to see that, at the limit of large sample size, the proposed approach can achieve perfect calibration with prediction accuracy close to the oracle.

4. The fourth concern is about the *generality of the results*. The experiments are conducted on 3 data sets, from which two (CIFAR10 and CIFAR100) are similar in nature. Since this paper is mostly empirical, it is most important to communicate to the reader that the proposed method is preferred on large-scale data sets. One such data set is ImageNet.

5. Lastly, the training time of the proposed method is 3X to 4X longer than the training time of a standard CE model. Here, it would be essential to compare the training time with the other baseline methods to have a better perspective.

---

> ### Author Response · Authors · 2023-07-06
> **Response to the review (1/2)**
>
> Thank you for reviewing our paper and for providing a specific list of points to discuss further. We provide responses to the individual points below. We especially appreciate that you find the idea of using meta-learning to improve calibration interesting and also that you highlight that our framework leads to impressive calibration.
>
> **(1) Comparisons: Meta-learning:** We kindly point out that the reviewer is mistaken to believe that Muller et al. “combine meta-learning with label-smoothing”. In fact, Muller et al. do not touch meta-learning at all. Applying meta-learning to solve calibration is the key high-level novelty of our work, which we implement specifically by our improved DECE objective, and our matrix label-smoothing (LS) representation. We build on Muller et al.’s observation that LS can affect calibration, but we exploit meta-learning to tune a LS matrix (cf: Muller’s LS scalar) to optimise the DECE meta-objective. Muller et al. do not impinge on this novelty as it does not touch meta-learning. Muller’s non-meta-learning LS method is already quantitatively compared throughout under the title “LS”, and we improve on their results substantially.
>
> **(1) Comparison: SB-ECE (Karandikar et al.):** We now include the comparison of error rates in addition to ECE in Table 11 and 12 in the appendix of the revised paper. Generally we see SB-ECE and SAvUC do not help as much for calibration when using the evaluation protocol from Mukhoti et al. The errors remain similar to Meta-Calibration and the other baselines, although instabilities can occur that lead to noticeably worse errors. When using SB-ECE and SAvUC within the outer loop loss they do not lead to as large improvements as our proposed meta-objective, but they achieve very similar accuracy.
>
> **(2) Accuracy:** Overall the accuracy of our approach is comparable to that of the other baselines (even if it is marginally worse in some cases) and the calibration itself is significantly better. There is a small number of cases when both ECE and accuracy are comparable to our approach, but their number is relatively small. The overall trend is that we obtain significantly better calibration at almost the same accuracy.
>
> **(3) Accuracy vs calibration trade-off:** Our main observation from the results is that the accuracy of Meta-Calibration is comparable while the calibration sees a large improvement. In this sense the trade-off is not a concern because we do not need to trade-off a noticeable amount of accuracy for better calibration with our method. To study this further we have now done analysis with simulated data as requested. We are grateful the reviewer has suggested this idea as it is a very valuable suggestion, and we now include it in the main part of the paper (within 4.2 Further analysis).
>
> More specifically the experiment with simulated data consists of 1) sampling parameters of a binary logistic regression model (oracle) over 2D data, 2) sampling the label for each data point from a binomial distribution with a class probability given by the oracle. 3) then we use the sampled data to train (i) a vanilla logistic regression model, (ii) label-smoothing logistic regression model and (iii) a meta-calibration logistic regression model.  As the reviewer suggests, this enables us to compare the best-case classifier accuracy and calibration with the results of learned models. Further details are provided in the revised version of the paper.
>
> The results in Table 7 in the revised paper confirm that Meta-Calibration matches both the accuracy and ECE of the oracle, obtaining close to the best possible calibration. Analogous vanilla and label smoothing (LS) logistic regression training have a significantly larger ECE. ECE of Meta-Calibration and Oracle is close to 0, but not precisely 0 due to sampling effects (i.e.: because we do not use an infinite amount of data). We also provide the summary table here. Figure 7 in the revised paper provides additional visualization of how Meta-Calibration compares with the oracle, simple cross-entropy as well as LS training.
>
> | Metric | Oracle | Cross-Entropy | LS | MC (Ours) |
> |--------|:------:|:-------------:|:---------:|:---------:|
> | Accuracy | 87.62 $\pm$ 1.87 | 87.55 $\pm$ 1.86 | 87.57 $\pm$ 1.87 | 87.55 $\pm$ 1.81 |
> | ECE  | 1.38 $\pm$ 0.32 | 9.81 $\pm$ 1.14 | 3.61 $\pm$ 0.43 | 1.40 $\pm$ 0.19 |

---

> ### Author Response · Authors · 2023-07-06
> **Response to the review (2/2)**
>
> **(4) Generality of the results:** We add a further dataset to our experiments to have a larger amount of datasets. Our Meta-Calibration leads to excellent calibration and comparable accuracy also on this further dataset. With the newly added scenario we cover 10 scenarios (CIFAR-10 with 4 models, CIFAR-100 with 4 models, SVHN with 1 model and 20 Newsgroups with 1 model), so a significant amount of scenarios is included. We have updated the tables in our paper to include this dataset (Tables 1-4 and Table 8).
>
> With regard to large scale datasets (ImageNet) we note that training models on ImageNet takes an impractical amount of time, especially when evaluating many approaches (6 baselines + our approach) and doing multiple repetitions to estimate the level of variability in the results. Further, meta-learning also adds to the cost of training. It is likely that with future developments in faster underpinning meta-learning, using large-scale datasets would be easier. In the meantime we believe our approach is already of interest to the ML community and will inspire further work on the important challenge of uncertainty calibration.
>
> **(5) Training time:** The other baselines have roughly similar runtime as CE because they only add further regularization or modify the loss function. For a fair comparison we have decided to train the base models using the same number of iterations for all approaches, so Meta-Calibration then takes longer due to the steps needed for training the hyperparameters. However, we note that if excellent calibration is key, we believe the additional time is acceptable. The training is a one-time cost, but calibration is important once the model is deployed and reused many times.

---

> > ### Comment · Reviewer_4bkh · 2023-07-27
> > **Thank you**
> >
> > I very much appreciate the detailed feedback that addressed most of my concerns. In particular, it is reassuring that the proposed method achieves nearly oracle performance in the new synthetic experiment.
> >
> > I believe including results on a larger data set (even on mini ImageNet) could be a plus, but I leave this decision to the AC.

---

### Review · Reviewer_8KmU · 2023-06-25

**Summary Of Contributions:**

The authors tackle two key challenges encountered during model training: 1) the non-differentiable nature of the calibration metric and 2) the overfitting of the calibration error to the training set.

In response to the first challenge, they devise a novel solution: DECE, a differentiable approximation to Expected Calibration Error (ECE). This approach facilitates direct calibration optimization during training.

To address the second challenge, the authors propose a meta-learning framework that uses DECE to optimize for calibration with a held-out calibration set.

They evaluate their methods and show that their approach yields well-calibrated, high-accuracy models that are competitive with existing methods on various benchmarks and architectures.

**Audience:**

Yes

**Claims And Evidence:**

Yes

**Requested Changes:**

* Comparison between interleaved training [1] and meta-calibration.
* Showing ImageNet result or NLP classification tasks would strengthen the results.
* Karandikar et al. [1] shows their results under data shift senarios (i.e., CIFAR-10C and CIFAR-100C). Does the proposed method still outperform other baselines under data shift senarios?

[1] Karandikar et al., Soft Calibration Objectives for Neural Networks, 35th Conference on Neural Information Processing Systems (NeurIPS 2021).


**Strengths And Weaknesses:**

Strengths

* Overall the paper is well written and easy to follow.
* The authors propose DECE to soften the ECE metric in order to make a differentiable calibration objectives. Although, this method is similar to Soft-binned ECE (SB-ECE) [1], the paper provides detailed comparison to SB-ECE in Appendix C.
* Calibration during training is particularly challenging as the overfitting is a significant challenge to solve this task (Figure 1). The paper proposes a meta-learning based method (i.e., meta-calibration) to solve this issue.

Weakness

* Lack of calibration results on large scale datasets such as ImageNet.
* The meta-calibration is proposed to solve the overfitting problem of trainset ECE. However, the paper does not compare this method to previously proposed solutions: Weighting method from [2] and interleaved training from [1]. Especially, the comparison between interleaved training vs. meta-calibration seems crucial since both method requires a calibration set, which takes about 10% of the training set dedicated to minimizing the calibration objective loss. What benefit would meta-calibration have over interleaved training?

[1] Karandikar et al., Soft Calibration Objectives for Neural Networks, 35th Conference on Neural Information Processing Systems (NeurIPS 2021).

[2] Kumar et al., Trainable Calibration Measures For Neural Networks From Kernel Mean Embeddings, Proceedings of the 35th International Conference on Machine Learning, Stockholm, Sweden, PMLR 80, 2018.

---

> ### Author Response · Authors · 2023-07-06
> **Response to the review**
>
> Thank you for reviewing our paper and for asking the questions. We provide answers to the questions below. We especially appreciate that you recognize the benefits of our framework for addressing overfitting of calibration and that you also highlight it leads to competitive well-calibrated, high-accuracy models.
>
> **Interleaved training:** We have now implemented and evaluated interleaved training as described in (Karandikar et al., 2021). We report both test ECE and error (%) in Table 13 and 14 in the appendix of the revised paper. The results suggest that interleaved training generally leads to worse ECE than our approach, in most cases significantly worse. Interleaved training also leads to significantly worse error compared to our Meta-Calibration and the other baselines. We attribute the empirical benefits of Meta-Calibration to using the calibration objective for training only the label smoothing meta-parameters and also the specialised meta-training that uses an inner and outer loop.
>
> **Comparison with the weighting method from Kumar et al.:** We already compared this method in our experiments. Please see Tables 1-4, etc., where we refer to Kumar et al. as MMCE. Our Meta-Calibration approach significantly outperforms this method.
>
> **Results on a NLP task:** One of our setups (20 Newsgroups) is a text classification task (NLP) so Meta-Calibration is already evaluated on NLP.
>
> **Results on large scale datasets (ImageNet):** Thanks for the suggestion, but we note that training models on ImageNet takes an impractical amount of time, especially when evaluating many approaches (6 baselines + our approach) and doing multiple repetitions to estimate the level of variability in the results. Further, meta-learning also adds to the cost of training. It is likely that with future developments in faster underpinning meta-learning, using large scale datasets would be easier. In the meantime we believe our approach is already of interest to the ML community and will inspire further work on the important challenge of uncertainty calibration.
>
> **Data shift scenarios:** We have now evaluated Meta-Calibration and the other baselines also in a cross-domain setup. We include the results in Table 9 and 10 in the appendix of the revised paper (section C Cross-domain evaluation). Meta-Calibration is still helpful in this case, but not to as large an extent as previously. Studying how to more successfully exploit meta-learning for calibration in cross-domain scenarios is an interesting research question that the ML community can focus on in the future.

---

### Review · Reviewer_5hfA · 2023-06-26

**Summary Of Contributions:**

This paper addresses the problem of calibration in deep learning. Modern deep learning can have discrepancy between confidence outputs (usually the value of softmax) and its local probability of success. The popular ECE is non-differentiable and cannot be used for gradient-based optimisation. Hence the proposal of a differentiable alternative. Such an objective is then used in meta-learning. However, the results of this paper are incongruent with results in previous literature. Therefore, the conclusion of the paper is not entirely clear. For example, the results in Table 1 and Table 2 are worse than Guo et al 2017 that the paper claims to improve upon.

Summary of contributions:

  * Although the experimental results leave some major questions, a differentiable objective for calibration error is a valid contribution. Also the study of calibration as a meta-learning problem seems novel.


**Audience:**

Yes

**Broader Impact Concerns:**

The paper addresses the problem of model calibration, which would have a positive impact on sensitive applications of deep learning.

**Claims And Evidence:**

No

**Requested Changes:**

The most important concern is to address the ECE values in Table 1 and Table 2, see previous section.




**Strengths And Weaknesses:**

# Strengths:

* The method section and algorithm are well explained.
* A differentiable objective for calibration error is a novel contribution.
* The ablation experiment in Figure 2 shows that the new objective aligns well with the established ECE objective.


# Weaknesses:

* Table 1 and 2 are confusing when comparing Guo et al results. The Guo et al. paper from 2017 shows ECE on CIFAR10 and 100 of 0.6% and 0.96%, respectively. However, the current results in this paper seem to be of higher/worse value, at 0.94% and 2.8%. Can this difference be explained?

* The discussion section mentions an increase in complexity and computation for the proposed method. Could this be quantified and compared? Does the increase in computation outweigh the benefits of ‘better’ calibration?


### Minor question:

* Figure 1 shows a discrepancy between ECE on train and validation set. However, this divergence only seems to happen on epoch 150, what makes this point so characteristic?
* On the reliability analysis in Figure 6: what information does this figure provide over the usual ECE metric? I have the impression that ‘the percentage of samples with various confidence levels‘ is merely a function of the dataset. When a calibrated model achieves 90+% accuracy on an easy dataset, then a well calibrated model would by definition have many high confidence predictions. Likewise for a ‘hard dataset’ one would expect even a well calibrated model to have many low confidence outputs.

---

> ### Author Response · Authors · 2023-07-06
> **Response to the review**
>
> Thank you for reviewing our paper and for asking the questions. We provide answers to the questions below. We especially appreciate that you recognize the novelty of studying calibration as a meta-learning problem.
>
> **ECE results discrepancy compared to Guo et al.:** The results are not supposed to be exactly comparable with Guo et al. The difference in the results is because (1) We follow the specific experimental setup of the more recent paper Mukhoti et al. (NeurIPS’20) whose codebase we extend. Since the training protocol is different (learning rate schedule, etc.) the results are not expected to match those in Guo et al. (2) We report the mean over multiple runs, unlike both these prior works that report only a single run.
>
> **Computation:** Increase in computation is quantified and reported in Table 8 in Appendix B. If excellent calibration is key, we believe the additional time is acceptable. The training is a one-time cost, but calibration is important once the model is deployed and reused many times for inference.
>
> **Learning curve behaviour:** Learning rate is decreased at epochs 150 and 250, so this leads to changes in behaviour at these points.
>
> **Reliability analysis:** The ECE metric is a distillation of the reliability diagram into a single scalar. Reporting the ECE alone would quantify the magnitude of the miscalibration, but does not show the *way in which a classifier is miscalibrated* – e.g., are there more high-confidence predictions than appropriate, or more low-confidence predictions than appropriate? The reliability diagram gives exactly this additional information.

---

### Author Response · Authors · 2023-07-06
**Updated version of the paper**

We thank all reviewers for reviewing our paper and providing valuable suggestions for improving it. In particular we appreciate that the reviewers recognize the novelty of studying calibration as a meta-learning problem (reviewer 5hfA), highlighting our solution leads to competitive well-calibrated, high-accuracy models (reviewer 8KmU) and that the idea of using meta-learning to improve calibration is interesting and leads to impressive calibration (reviewer 4bkh).

We have improved our paper based on suggestions of the reviewers and we highlight the revised parts of our paper in red. In particular, we add the following:
* Results on a further dataset: SVHN (main part)
* Analysis of accuracy vs calibration using simulated data (main part)
* Cross-domain evaluation (appendix)
* Test error rates for SB-ECE (appendix)
* Interleaved training (appendix)

We also provide detailed responses to the questions from each reviewer. We believe our responses as well as the improvements to the paper give good justifications to recommend acceptance for our paper.

---

### Decision · Action_Editors · 2023-08-07

**Recommendation:** Accept with minor revision

**Comment:**

Reviewers think the novelty is minor, but they commend the paper as of high clarity. The main concerns raised by them are:
1. Large-scale dataset experiments are missing;
2. Missing some comparisons with soft-binned ECE and interleaved training which are closely related to the proposed approach;
3. Worse ECE results compared with Guo et al. (2017).

In revision the authors added more experiments and baselines as suggested by point 2 above. The reviewers are satisfied for most of the additional results, however the 3rd point above still remain unaddressed.

The authors replied Concern 3 as "we followed the setting of Mukhoti et al. (2020) and used their code to build our codebase". While in different settings the ECE results for the calibration results may be different, the authors need to clarify in the paper on exactly how different the settings are. For example, if the network architecture and data augmentation procedure used in Guo et al. (2017) are exactly the same as what were used in this paper, then I personally don't think "using different learning rates" is a valid reason for not attaining Guo et al. (2017) performance, unless the authors have evidence to claim that Guo et al. results are not reproducible.

I suggest the authors to revise accordingly -- I think the acceptance should be conditioned on clarifying the above point.

**Audience:**

Researchers interested in uncertainty estimation in neural networks as well as meta-learning & distribution shift.

**Claims And Evidence:**

This paper proposed a differentiable version of expected calibration error (ECE) by extending the soft-binned ECE from Karandikar et al. (2021). The authors then used the proposed differentiable ECE in a meta-learning procedure to address potential issue of worse trade-offs between accuracy and calibration. Experiments showed the proposed approach is competitive, outperforming the baselines included in the paper.

---

> ### Author Response · Authors · 2023-08-15
> **Updates to the paper**
>
> Thank you very much for handling our paper, and we highly appreciate the decision to accept our paper with minor revision. We have now added a new section to the appendix of our paper where we discuss in detail differences compared to Guo et al. The key points that we mention include the following:
>
> 1) The gap is less when comparing exactly the same models (i.e. compare the result in Guo et al. for the standard ResNet110 that we evaluate rather than their result for stochastic depth SD-ResNet110 that gave the values mentioned by the reviewer),
> 2) The fact that the results in Guo et al. obtain stronger calibration while having significantly worse error rates than Mukhoti et al.,
> 3) Details of probable differences in the training procedure - primarily different number of epochs, learning rate annealing schedule, weight decay, minibatch size and the use of Nesterov momentum (only probable differences due to the level of information provided by Guo et al.),
> 4) We show that across various reasonable hyperparameters based on information in the paper and demo code from Guo et al., their results are not straightforward to reproduce. We include a table with detailed evaluation in the new section to support our claims about lack of reproducibility.
>
> Overall we believe it is fair to follow the setup from Mukhoti et al. because they provide a complete and reproducible codebase, full details of their experiments and also strong overall results, especially in terms of error rate. Please let us know if you have any questions and we will be happy to answer.